# Validation of an Algorithm for Measurement of Sedentary Behaviour in Community-Dwelling Older Adults

**DOI:** 10.3390/s23104605

**Published:** 2023-05-09

**Authors:** Khalid Abdul Jabbar, Javad Sarvestan, Rana Zia Ur Rehman, Sue Lord, Ngaire Kerse, Ruth Teh, Silvia Del Din

**Affiliations:** 1School of Population Health, Faculty of Medical and Health Sciences, University of Auckland, Auckland 1023, New Zealand; khalid.abdul-jabbar@auckland.ac.nz (K.A.J.); r.teh@auckland.ac.nz (R.T.); 2Translational and Clinical Research Institute, Faculty of Medical Sciences, Newcastle University, Newcastle upon Tyne NE2 4HH, UK; javad.sarvestan@newcastle.ac.uk (J.S.); or rrehman5@its.jnj.com (R.Z.U.R.); silvia.del-din@newcastle.ac.uk (S.D.D.); 3Janssen Research & Development, High Wycombe HP12 4EG, UK; 4School of Clinical Sciences, Auckland University of Technology, Auckland 1010, New Zealand; sue.lord@aut.ac.nz; 5National Institute for Health and Care Research (NIHR), Newcastle Biomedical Research Centre (BRC), Newcastle University, The Newcastle upon Tyne Hospitals NHS Foundation Trust, Newcastle upon Tyne NE2 4HH, UK

**Keywords:** real-world, sedentary behaviour, validation, older adults, wearable device, digital health

## Abstract

Accurate measurement of sedentary behaviour in older adults is informative and relevant. Yet, activities such as sitting are not accurately distinguished from non-sedentary activities (e.g., upright activities), especially in real-world conditions. This study examines the accuracy of a novel algorithm to identify sitting, lying, and upright activities in community-dwelling older people in real-world conditions. Eighteen older adults wore a single triaxial accelerometer with an onboard triaxial gyroscope on their lower back and performed a range of scripted and non-scripted activities in their homes/retirement villages whilst being videoed. A novel algorithm was developed to identify sitting, lying, and upright activities. The algorithm’s sensitivity, specificity, positive predictive value, and negative predictive value for identifying scripted sitting activities ranged from 76.9% to 94.8%. For scripted lying activities: 70.4% to 95.7%. For scripted upright activities: 75.9% to 93.1%. For non-scripted sitting activities: 92.3% to 99.5%. No non-scripted lying activities were captured. For non-scripted upright activities: 94.3% to 99.5%. The algorithm could, at worst, overestimate or underestimate sedentary behaviour bouts by ±40 s, which is within a 5% error for sedentary behaviour bouts. These results indicate good to excellent agreement for the novel algorithm, providing a valid measure of sedentary behaviour in community-dwelling older adults.

## 1. Introduction

Sedentary behaviour (SB) is defined as ”any waking activity characterised by an energy expenditure ≤ 1.5 metabolic equivalents (METs), whilst in a sitting, reclining or lying posture” [1]. SB is distinct from physical inactivity and is differentially associated with health risks. High levels of SB are unfavourably related to cognitive function, depression, functional status, and disability in older adults [2,3,4]. For example, increased sitting duration, especially if associated with more screen time (e.g., watching television or use of mobile phones but not computer or internet use time [2]), are detrimental to sleep health [5,6] and social connectedness [7] which could increase the risk of disability, loneliness, and depression in adults [8]. Traditional methods of quantifying SB (e.g., questionnaires and diaries) have the potential for inaccuracy and inherent bias (e.g., recall bias) [9]; thus, the use of wearable devices (wearables) to objectively quantify SB is a welcome advancement in the field.

There has been a dramatic increase in studies that employ wearables to investigate SB, including some large-scale longitudinal population studies [10,11]. Wearable devices allow objective yet continuous and unobtrusive tracking of movement and posture and provide refined and accurate data on sedentary activities [12]. However, only a limited number of studies have investigated the validity of algorithms related to SB using accelerometry in the older populations who are the most sedentary amongst the age groups [13,14]. Out of 15 of these reported studies, only 7 investigated the accuracy of SB in real-world environments [14]. Studies employing machine learning techniques fared better than those relying on other techniques in detecting real-world SB, but more rigorous field-based research is still warranted [14,15].

Furthermore, algorithms designed to identify SB are limited in scope and, overall, perform inconsistently [16]. For example, discerning sitting from standing is problematic [17,18]. Studies that employ multiple sensor configurations report better accuracy [19,20,21], but such configurations increase the wearability burden especially if used for longer periods of time. Others that employ a single wearable device usually use the thigh as the preferred site (e.g., [22,23]) given that the wrist is preferred for monitoring physical activities [24]). However, a single wearable limits the ability to accurately distinguish sedentary activities such as sitting from lying (e.g., afternoon napping, which is common in older adults) both of which are important in the case of older adults [17,25]. In addition, algorithms based on machine learning and artificial intelligence techniques that solely rely on fixed cut-off points (primarily based on activity counts/step counts or METs) to classify SB are usually difficult to generalise in a population that the algorithm was not trained for, [26] limiting widespread utility for these algorithms.

Accurate and reliable measurement of sedentary behaviour in older adults is informative and relevant and will allow us to plan appropriate intervention strategies. A single open-source, accelerometer-based wearable—the Axivity monitor—attached to the lower back has been recently validated to detect a comprehensive battery of real-world gait characteristics in older adults [27,28]. Whether the same configuration can be used to detect SB remains to be investigated. In this study, we developed an algorithm that uses specific characteristics of the participants to detect SB.

Thus, the main objective of this study was to validate the performance of a customised algorithm based on a single wearable device placed on the L5 position of the lower back (to increase usability and acceptability) to identify SB and to discriminate its domains—sitting versus lying versus upright, in community-dwelling older people aged 75 years and above, in real-world conditions.

## 2. Materials and Methods

### 2.1. Participants

This study was embedded in the Ageing Well Through Eating, Sleeping, Socialising and Mobile (AWESSoM) study [10]. Older adults participating in AWESSoM were invited to take part, alongside participants who met the following inclusion criteria but did not enrol in AWESSoM. Inclusion criteria were (1) age of 75 years or over; (2) able to ambulate a minimum of 15 m independently, with or without walking aids; (3) able to stand, with or without walking aids, for a minimum of 60 s. Exclusion criteria were (1) any significant medical, orthopaedic, or neurological conditions that would contraindicate normal activity; (2) allergy to surgical adhesive tape. All subjects gave their informed consent for inclusion before they participated in this study. This study was conducted in accordance with the Declaration of Helsinki, and the protocol was approved by the New Zealand Ministry of Health and Disability Ethics Committee (2021 AM 9955).

### 2.2. Experimental Protocol

The Axivity monitor (AX6) is a wearable device incorporating a triaxial accelerometer and gyroscope, with a sampling frequency of 100 Hz, accelerometer range: ±8 g, and gyroscope range: 2000 degrees per second (dps). It is firmly established as a robust single-wearable device, used extensively to measure continuous real-world activity across the age range (e.g., younger adults [29], older adults [27]). For this study, we secured the AX6 onto the lower back at the fifth lumbar vertebrae of each participant, using a hydrogel adhesive, covered with a surgical-grade adhesive dressing (OPSITE Flexifix™ or Hypafix™, Smith+Nephew Ltd., Watford, UK). A handheld tablet (Galaxy Tab A, SM-P555, Samsung, sampling frequency: 30 FPS (frames per seconds), resolution: 1280 × 720) was used for videoing all movements of the participants. The AX6 and the handheld tablet were time synchronised using the network time (https://nist.time.gov/) via a laptop connected to the internet. This was performed with the respective USB cables that came with the devices connected to the laptop.

### 2.3. Procedure

Both scripted and non-scripted sedentary activities (sitting and lying bouts) were defined based on prior research [30,31,32]. Participants undertook these activities in their own homes or retirement villages (Appendix A). Both tasks were video recorded by a research assistant using a handheld tablet, and was restricted to the trunk and lower limbs, and all recognisable features (e.g., facial) were avoided.

#### 2.3.1. Scripted Activities

To indicate the start of the scripted activity and for synchronisation purposes, the AX6 was tapped by the research assistant three times at approximately one-second intervals. Participants then completed the following activities sequentially: (a) from a standing position, sit on a lounge/sofa chair for (approximately) one minute; (b) stand up and walk at their comfortable pace (with or without walking aids) to their dining area and sit on their dining chair for one minute; (c) stand up and walk to their bedroom and lie on their back on their bed for one minute; (d) sit up on the edge of their bed for approximately three seconds, then stand up (with or without support) beside their bed for one minute; (e) walk to their dining area; (f) when they are about to reach their dining area, they are instructed to return back to their bedroom; (g) when they are about to reach their bedroom, they are instructed to return to their lounge area; (h) sit down on their lounge chair for one minute; (i) to stand up and stand still for one minute. The AX6 was then tapped by the research assistant three times at approximately one-second intervals. This completes the scripted activity. Participants were requested to rest before executing the non-scripted activities.

#### 2.3.2. Non-Scripted Activities

Participants were then instructed to continue their activities as normal for a duration of up to eight minutes. They were requested to avoid sitting or lying for too long during this period. To indicate the end of the non-scripted activity, they were instructed to return to their lounge area and sit on their lounge chair for approximately 60 s and thereafter stand up. The AX6 was then tapped by the research assistant three times at approximately one-second intervals, for synchronisation purposes. This completes the non-scripted activity.

#### 2.3.3. Data Management

Data from the wearable device were downloaded to a computer using the OmGui software (Version 1.0.0.43, Open Movement, Newcastle, UK). Selected data based on the start and the end timing of the scripted and unscripted activities, respectively, were exported as raw comma-separated values (CSV) files, with the timestamps option as ”Fractional days (MATLAB)”. The data were then resampled at 100 Hz and linearly interpolated (piecewise cubic hermite interpolating polynomial [pchip]) in MATLAB (R2022a) to address the issue of real-time clock drift within the AX6. The video and the resampled data were frame synchronised using the ELAN software (Version 6.2, Nijmegen: Max Planck Institute for Psycholinguistics, The Language Archive) by identifying the exact start frame of the first tap on the AX6 (see Section 2.3.1 and Section 2.3.2). Participants’ activities (see Table 1) were coded based on the video recordings by the observer (KAJ). From the coded information, the duration of sitting, lying, and upright activities were calculated based on the start and end frame of each activity.

#### 2.3.4. Algorithm Implementation

The algorithm used in this study is described in Algorithm 1 (see Figure 1 for flow). The key phases of the algorithm are data preparation, classification, and detection. The raw triaxial accelerometry data were first removed of their offset (mean accelerations) and thereafter passed through a 2nd order low-pass Butterworth two-pass filter with a cut-off frequency of 17 Hz [33]. A moving window of 0.1 s (i.e., 10 data samples) was then used [34] to identify upright activities based on the likelihood of whether the participant was a more “upright” (likely to spend more time in upright activities) or less “upright” (likely to spend more time in sitting and lying activities) candidate. Mediolateral and anteroposterior tilt thresholds were estimated based on upright versus non-upright postures. The vertical tilt threshold was estimated from earlier studies on gait and sit-to-stand movements in older adults [35,36,37,38]. The start and end of each potential upright bout were then identified and stored. Thereafter, appropriate thresholds were applied to the filtered anterior–posterior tilt angles to confirm the upright bouts. The next part of the algorithm identified the activities between any two consecutive upright bouts. For this, we assumed that the filtered mean anterior–posterior tilt angles of any lying activities should be at least 2.5 times lower than those of the preceding upright bout. If this was not true, then we checked whether the former was less than the filtered mean anterior–posterior tilt angles of the preceding upright bout. If true, then the current non-upright bout is likely to be a sitting bout, else it is likely an upright bout.
**Algorithm 1** Pseudocode for sitting, lying, and upright bouts**Data**: *acc* = [*ax*, *ay*, *az*] ^1^ *ax_filtered_ ← butterworth* (*ax*, *order* = 2, *cutoff* = 17 Hz) *ay_filtered_ ← butterworth* (*ay*, *order* = 2, *cutoff* = 17 Hz) *az_filtered_ ← butterworth* (*az*, *order* = 2, *cutoff* = 17 Hz) **for**
*(every 0.1 s)* *std_ax_filtered_ ← stdev(ax_filtered_)* *std_ay_filtered_ ← stdev(ay_filtered_)* *std_az_filtered_ ← stdev(az_filtered_)* *tilt_angle_VT ←* $arccos(\frac{ax}{\sqrt{ax^{2}+ay^{2}+az^{2}}})$*·(*$\frac{180}{\pi }$*)* *tilt_angle_ML ←* $arccos(\frac{ay}{\sqrt{ax^{2}+ay^{2}+az^{2}}})$*·(*$\frac{180}{\pi }$*)* *tilt_angle_AP ←* $arccos(\frac{az}{\sqrt{ax^{2}+ay^{2}+az^{2}}})$*·(*$\frac{180}{\pi }$*)* **end** *std_sum ← std_ax_filtered_ + std_ay_filtered_ + std_az_filtered_* *std_sum_filtered_ ← butterworth* (*std_sum*, *order* = 2, *cutoff* = 1 Hz) *tilt_angle_VT_filtered_ ← butterworth* (*tilt_angle_VT*, *order* = 2, *cutoff* = 0.25 Hz) *tilt_angle_ML_filtered_ ← butterworth* (*tilt_angle_ML*, *order* = 2, *cutoff* = 0.25 Hz) *tilt_angle_AP_filtered_ ← butterworth* (*tilt_angle_AP*, *order* = 2, *cutoff* = 0.25 Hz) **for**
*(every 0.1 s)*    *create empty array to store upright movement* **end** **if**
*ceiling*(*mean*(*tilt_angle_VT_filtered_) ≥ 150 ^2^*    **if**
*ceiling*(*mean*(*tilt_angle_ML_filtered_) ≥ 90 ^2^*       **if**
*ceiling*(*mean*(*tilt_angle_AP_filtered_) ≥ 90*
^2^
         **for**
*(every 0.1 s)*           **if**
*std_sum_filtered_ ≥ mean(std_sum_filtered_)*              *assign 1 to the array ^3^*            **end**         **end**       **end**    **end** **else if**
*tilt_angle_VT_filtered_ ≥ 140 ^2^*
**and**
*tilt_angle_AP_filtered_ ≥ 75 ^2^*     **for**
*(every 0.1 s)*       *assign 1 to the array*
^3^     **end** **end** **find**
*start_frame and end_frame of potential upright bouts and store in an array* **Result:**
*MoveArray [start_frame, end_frame] ^4^* **for**
*every two consecutive potential upright bouts*    **if**
*mean*(*tilt_angle_AP_filtered_) of current potential upright bout < 40 ^5^*       *label current potential upright bout as “Lying”*    **else if**
*mean*(*tilt_angle_AP_filtered_) of current potential upright bout < 80 ^5^*       *label current potential upright bout as “Sitting”*    **else**       *label current potential upright bout as “Upright”*    **end**    **if**
*mean*(*tilt_angle_AP_filtered_) of current non-upright bout < mean*(*tilt_angle_AP_filtered_)/2.5 of preceding upright bout*       *label current non-upright bout as “Lying”*    **else if**
*mean*(*tilt_angle_AP_filtered_) of current non-upright bout < mean*(*tilt_angle_AP_filtered_) of preceding upright bout*       *label current non-upright bout as “Sitting”*    **else**
       *label current non-upright bout as “Upright”*    **end** **end** **Result:**
*Data_Label =* [array of labelled bouts]

In Algorithm 1, ^1^—*ax*—vertical axis, *ay*—mediolateral axis, *az*—anterior–posterior axis; ^2^—these thresholds were estimated based on earlier studies [35,36,37,38]. ^3^—“1” indicates “upright”. ^4^—Array with start and end frame number of “upright” movement. Note that the end_frame of the current frame to the start of the next start_frame was considered as “non-upright” bout. ^5^—These thresholds were estimated based on the whole dataset.

#### 2.3.5. Data Analysis

To determine *inter-rater reliability*, ten video recordings were randomly selected and the start and end frame of sitting, lying, and upright activities (see Table 1 for definitions) of both scripted and non-scripted activities were independently annotated using the ELAN software by two investigators (SL and KAJ). The results were presented as intra-class correlation (two-way random, absolute agreement) [ICC_(2,1)_]. Linear relationships of the duration of activities between the algorithm and the observer were also investigated using ICC_(2,1)_ to establish levels of agreement. *Criterion validity* between the analysed accelerometer data and the corresponding video observations (considered the “gold standard”) of time resolution of 0.01 s (based on the 100 Hz sampling frequency of the AX6) was assessed based on the sensitivity, specificity, positive predictive value (PPV), and negative predictive value (NPV). These measures are described as follows:(1)$$\mathrm{Sensitivity}=\frac{TP}{TP+FN},$$
(2)$$\mathrm{Specificity}=\frac{TN}{TN+FP},$$
(3)$$\mathrm{Positive predictive value (PPV)}=\frac{TP}{TP+FP},$$
(4)$$\mathrm{Negative predictive value (NPV)}=\frac{TN}{TN+FN},$$

True positives, true negatives, false positives, and false negatives are described in Figure 2 below. Sensitivity describes how well the algorithm correctly identifies each observed category of activities (i.e., sitting, lying, and upright). Specificity describes how well the algorithm correctly identifies the absence of each observed category of activities (i.e., not sitting, not lying, and not upright). PPV describes the probability that when the algorithm identifies an activity that was present, it is actually correct. NPV describes the probability that when the algorithm identifies an activity that was absent, it is actually correct.

Bland–Altman plots were used to investigate the limits of agreement between the total duration of each activity [39]. The absolute percentage error (*APE*) and the absolute error (*AE*) of each activity were calculated as the difference between the accelerometer and video observation duration divided by the video observation duration. Statistical and graphical analysis were performed in R Studio (Version 3.6.1).
(5)$$AE=\frac{1}{n}\sum_{t=1}^{n}\left|R_{t}-A_{t}\right|,$$
(6)$$APE=\frac{100\%}{n}\sum_{t=1}^{n}\left|\frac{R_{t}-A_{t}}{R_{t}}\right|,$$
where *R* is the duration of each individual activity based on the reference (video observation), *A* is the duration of each individual activity based on the algorithm, and *n* is the number of bouts.

## 3. Results

### 3.1. Participants

A total of twenty older adults participated in this study. Of these, 19 were also part of the AWESSoM study. All participants completed both scripted and non-scripted activities. Data for two participants could not be processed because of synchronisation and scripted-task errors. The average ±SD age for the remaining 18 participants was 81.1 ± 6.2 years, and more than 60% were females (Table 2). A total of 293.27 min of sedentary behaviour (scripted—89.59 min, non-scripted—203.68 min) was analysed. No lying activities were captured during the non-scripted activities (Table 3).

### 3.2. Inter-Rater Reliability

Inter-rater reliability, ICC_(2,1),_ for both investigators (SL, KAJ) was calculated based on nine videos because one of the videos had synchronisation issues. The ICC_(2,1)_ for sitting, lying, and upright activities was 0.999, 0.985, and 0.999, respectively. The intra-class correlation between the algorithm and the observer was good to excellent for all activities. The ICC_(2,1)_ [scripted, non-scripted] for sitting activities was 0.888 and 0.981, for upright activities it was 0.946 and 0.997, and for scripted lying activities it was 0.858 [40] (Table 4).

### 3.3. Criterion Validity

Table 5 shows the sensitivity, specificity, PPV, and NPV for both scripted and non-scripted activities. Sensitivity, accuracy, and PPV for scripted activities were lower compared to non-scripted activities. Sensitivity was lowest (70.39%) for scripted lying activity and highest (96.94%) for non-scripted upright activity. Specificity was high (≥90%) for both scripted as well as non-scripted activities. The algorithm was able to correctly identify sitting, lying, and upright activities with a probability of ≥83% for scripted and ≥95% for non-scripted (based on PPV). It was able to identify non-sitting and non-upright non-scripted activities with probabilities above 90% (based on NPV). The algorithm showed a lower probability for scripted activities, especially for scripted upright activities (NPV—75.89%).

### 3.4. Limits of Agreement

Bland–Altman plots are shown in Figure 3. The absolute mean difference (bias) between the algorithm and the video annotation was less than 10 s for all activities (range: 0.75 s to 9.61 s). The algorithm overestimated scripted lying activities by 9.61 s. It underestimated sitting by more than 3 s in both scripted and non-scripted conditions. The limits of agreement were greatest for (individual) non-scripted sitting activities: −30.49 s to 23.27 s, and the lowest were for scripted upright activities: −12.73 s to 11.12 s. The absolute percentage errors were relatively low (<16.1%) for sitting and upright activities but not so for lying activities (22.4%) (Table 6).

## 4. Discussion

The main goal of this study was to validate the real-world performance of a novel customised algorithm for identifying SB (sitting, lying, and upright activities) in community-dwelling adults aged 75 years and over. The PPV (>80%) and NPV (>75%) indicated good agreement between the algorithm and video observations for all activities, although the algorithm generally fared better in non-scripted activities than scripted activities. The limits of agreement (Bland–Altman plots) suggested that the algorithm could, at worst, overestimate or underestimate sitting, lying, or upright activities by ±40 s, which is within a 5% error for the average duration of bouts for sitting [41], lying (daytime napping) [42], and upright [43] in generally healthy community-dwelling older adults.

The PPV for all three activities surpassed that reported by Dijkstra et al. [44] (Table 7) by at least 10%. Although the current algorithm performed better in detecting sitting and upright activities compared to those reported by Taylor et al., it compared unfavourably for lying activities [17]. This could be due to the difference in age. Taylor et al. investigated an older age group (88.1 ± 5.0 years) which included long-term care participants, and the duration and number of occurrences of lying activities were greater than in the current study [17]. Other notable differences between the present study and that of Taylor et al. were that their real-world tasks included lying as a prescribed activity. In addition, the sensor used in their study (DynaPort MoveMonitor, McRoberts, The Hague, the Netherlands) differed from ours. However, similar to our results, Taylor et al. also reported that their algorithm performed better in non-scripted activities than in scripted activities. This could be due to the lower number of transitions and longer duration of activities within the non-scripted activities compared to the scripted activities [17].

The current algorithm emphasises the posture of the trunk rather than the intensity of the movement (i.e., raw accelerometry data) and looks for SB between two identified upright activities. The algorithm uses accelerometry data from the whole dataset to estimate the mean tilt angles (vertical, anterior–posterior, medial–lateral), and based on these angles and their respective (fixed) thresholds, classifies the participant as “more likely to spend more time upright” or “less likely to spend more time upright”. If the participant is “more likely to spend more time upright”, it then uses the standard deviation of the triaxial acceleration (signal vector magnitude) [29] to classify the upright activity. Otherwise, it uses fixed thresholds to classify upright activities based on vertical and anterior–posterior tilt angles alone. Because the algorithm for the “more likely to spend more time upright” scenario does not incorporate postural information, it is more sensitive to gait but less so for upright standing activities, which is at times misclassified. The other issue with the current algorithm is that it overestimates lying durations. The algorithm includes postural transitions (i.e., stand to sit, sit to lie, lie to sit, and sit to stand) within the duration of lying activities, which thus inflates the actual durations of lying activities. All thresholds for this study were tuned to improve the detection of sitting activities rather than lying activities because we anticipate more bouts of sitting activities for this cohort of older adults. This could be a plausible reason why the current algorithm failed to perform well for detecting lying bouts when compared to earlier studies [17,18].

The configuration purposefully adopted in this study used only the lower back with a single wearable to minimise the wearability burden on its users. Although the main objective of this study was to quantify SB, information of PA that includes gait and turning is important for understanding change in functional decline in older adults. The lower back and the hip are recommended for gait-related activities as these locations are closest to the centre of gravity of the participants [45]. Even with this limitation, the current algorithm generally fared better than previously published algorithms with similar configurations. The key improvement in the current algorithm is the semi-adaptative approach to understanding the user. It tries to classify the user based on the amount of time they spent in upright activities and non-upright activities. This step helps the algorithm to use an appropriate threshold—standard deviation of accelerations versus tilt angles—to better identify upright bouts. Furthermore, the algorithm also uses the participant’s own postural information in estimating the thresholds for tilt angles rather than fixed thresholds to classify sitting and lying bouts.

### Study Limitations

We wish to acknowledge that our use of reference (video observation method), although considered as “gold standard”, is still prone to subjectivity. More costly but better alternatives, such as optical-based systems, are available. The current algorithm relied mainly on tilt angles and accelerometry data to classify the activities. It also used fixed thresholds (in addition to customised thresholds) to differentiate activities. Newer research-grade wearables have an in-built triaxial gyroscope that provides additional (trunk) rotational information of the participant. They may classify sedentary and non-sedentary activities better. Accurately identifying key events and postural transitions, such as the initiation of a gait and the start and end of a sit-to-stand transition, may inform us of the precise timing of when an activity ends and when a new activity begins. Furthermore, the duration of postural transitions, although they could be considerably much smaller compared to the duration of sitting or lying, were not identified as separate activities in this study. These factors have limited the ability and performance of the algorithm. Some newer algorithms incorporate machine learning and artificial intelligence to improve their accuracy [20,21], albeit they might lack the necessary generalisability to be adopted for a wider population. These algorithms could be used to estimate customised thresholds and use additional signal-related features to not only identify SB and PA, but also accurately classify the key postural transitions.

## 5. Conclusions

This study investigated the ability of a semi-personalised algorithm to identify SB and discriminate sitting, lying, and upright activities. This was conducted in real-world conditions with minimal experimental setup and constraints. The importance of measuring SB in addition to PA in older adults is well recognised, but accurate and reliable measurements in daily life are challenging. The current algorithm provides a valid measure for identifying SB in community-dwelling older adults in real-world conditions and this could provide researchers in this field with better and clearer understanding on how SB plays an important role in healthy ageing. However, the algorithm’s accuracy, especially for lying activities, could be improved if postural transitions were separately classified.

## Figures and Tables

**Figure 1 sensors-23-04605-f001:**
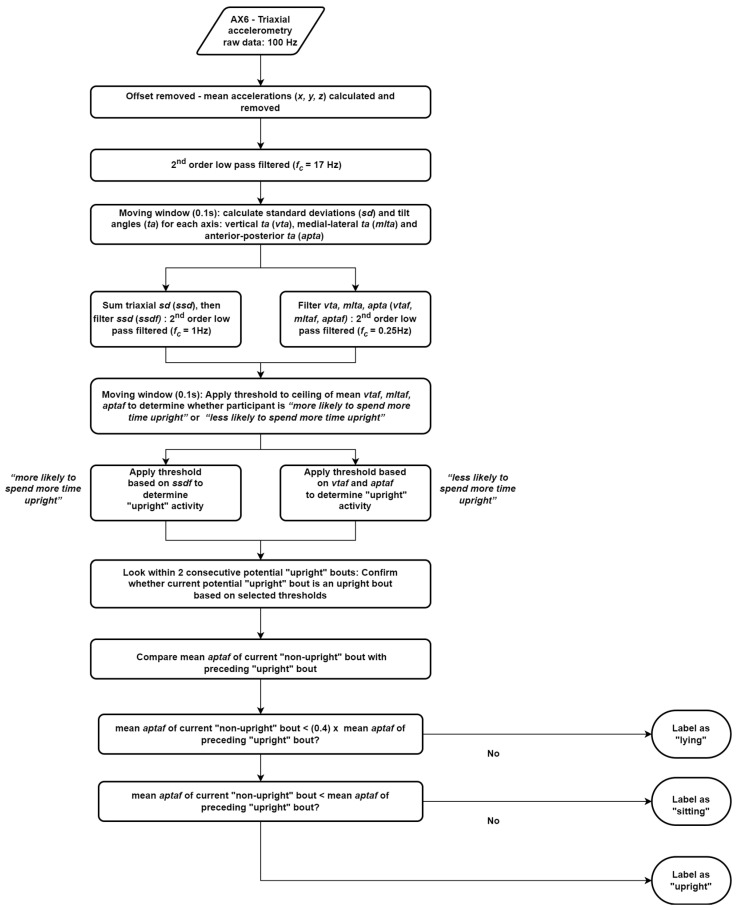
Flowchart showing key phases of the algorithm.

**Figure 2 sensors-23-04605-f002:**
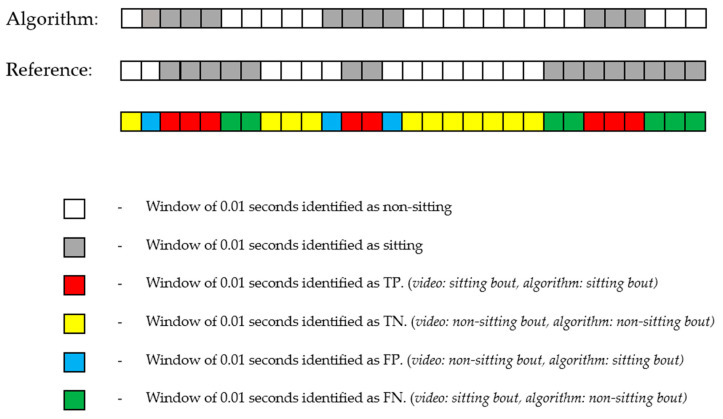
Example of sitting identification. Each square represents a window of 0.01 s. The reference here refers to the video observations.

**Figure 3 sensors-23-04605-f003:**
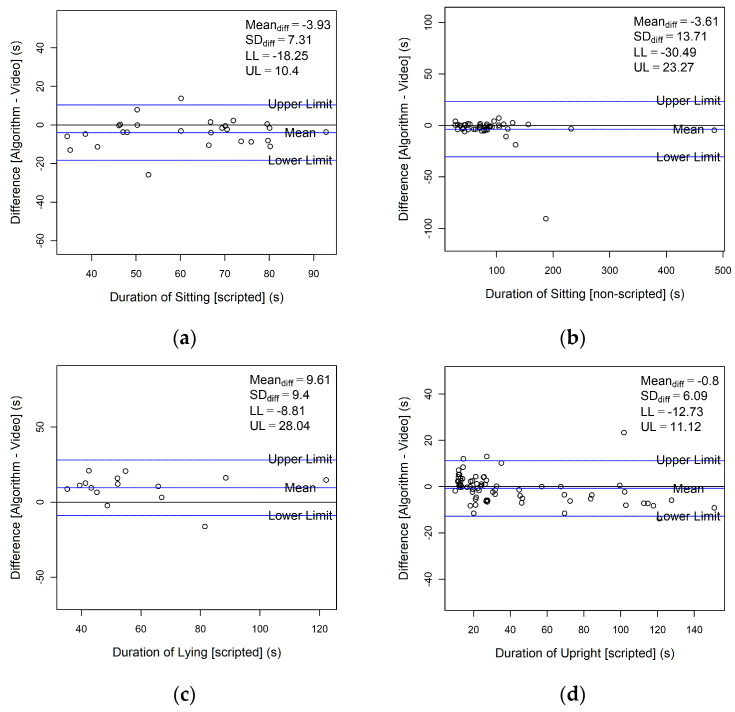
Bland–Altman plots of the duration of sitting, lying, and upright activities. Each point in the graph represents an individual’s activities. (**a**) Duration of scripted sitting bouts; (**b**) duration of non-scripted sitting bouts; (**c**) duration of scripted lying bouts; (**d**) duration of scripted upright bouts; (**e**) duration of non-scripted upright bouts.

**Table 1 sensors-23-04605-t001:** Definitions of SB and upright activities.

Event/Activity	Definitions
Sitting	- when the participant’s buttock is fully in contact with the seat of the chair/bed/stool (i.e., not on the ground). (adapted from [32])
Lying	- when the participant’s trunk and thigh are in a relatively horizontal posture with the back and stomach or side touching a horizontal underground. (adapted from [30,31,32])
Upright	- any activity undertaken with only the feet (and the use of assisted devices, if applicable) touching the ground. This would include standing without upper body movement, standing with upper body movement, walking, running, shuffling, stair climbing, stair descending. (adapted from [30,31,32])

**Table 2 sensors-23-04605-t002:** Characteristics of participants [Mean ± SD].

	Scripted (n = 18)	Non-Scripted (n = 17)
Age (yrs.)	81.1 ± 6.2	80.5 ± 5.9
Female	12 (66.7%)	11 (64.7%)
Weight (kg)	71.2 ± 13.1	72.2 ± 12.7
Height (cm)	163.1 ± 9.4	163.8 ± 9.2
BMI	26.6 ± 3.3	26.7 ± 3.3

**Table 3 sensors-23-04605-t003:** Total duration, average duration [Mean ± SD].

	Total Duration (in secs)				Average Duration ^1^ (in secs)	
Activity	Scripted	Non-Scripted	Overall	Scripted	Non-Scripted	Overall
Sitting	1654.8	4111.2	5766.0	61.3 ± 16.0	89.4 ± 72.5	79.0 ± 59.7
Lying	879.7	NA ^2^	879.7	58.6 ± 23.4	NA ^2^	58.6 ± 23.4
Upright	2840.9	8109.6	10,950.4	40.0 ± 35.6	180.2 ± 217.1	94.4 ± 153.3

^1^ Based on the number of activities. ^2^ No non-scripted lying activities were captured.

**Table 4 sensors-23-04605-t004:** ICC_(2,1)_ between the duration of activity [Mean ± SD] of video reference and algorithm.

	Video Reference (in secs)			Algorithm (in secs)			ICC_(2,1)_		
Activity	Scripted	Non-Scripted	Overall	Scripted	Non-Scripted	Overall	Scripted	Non-Scripted	Overall
Sitting	61.3 ± 16.0	89.4 ± 72.5	80.0 ± 59.7	57.4 ± 18.1	85.8 ± 70.4	75.3 ± 58.3	0.888	0.981	0.923
Lying	58.6 ± 23.4	NA	58.6 ± 23.4	68.3 ± 24.4	NA	68.3 ± 24.4	0.858	NA	0.858
Upright	37.4 ± 33.7	180.2 ± 217.1	102.4 ± 163.9	39.4 ± 32.3	183.0 ± 219.0	104.7 ± 165.1	0.946	0.997	0.997

**Table 5 sensors-23-04605-t005:** Sensitivity, specificity, positive predictive value [PPV], and negative predictive value [NPV]. Scripted activities (n = 18) and non-scripted activities (n = 17).

	Sensitivity (%)		Specificity (%)		PPV (%)		NPV (%)	
	Scripted	Non-Scripted	Scripted	Non-Scripted	Scripted	Non-Scripted	Scripted	Non-Scripted
Sitting	76.90	92.25	94.84	99.48	89.17	98.99	88.14	95.83
Lying	70.39	NA	95.71	NA	83.10	NA	91.51	NA
Upright	77.30	96.94	92.54	98.99	93.07	99.48	75.89	94.25

**Table 6 sensors-23-04605-t006:** Absolute percentage error (APE) and absolute error (AE) [Mean ± SD].

Activity	Scripted	Non-Scripted	Scripted	Non-Scripted
Sitting	10.8 ± 11.8	5.0 ± 7.6	5.9 ± 5.8	4.9 ± 13.3
Lying	22.4 ± 11.9	NA	12.1 ± 5.6	NA
Upright	16.1 ± 16.9	10.0 ± 13.5	4.5 ± 4.2	6.4 ± 14.8

**Table 7 sensors-23-04605-t007:** Comparison of PPV (%) of current algorithm (free-living data only).

Activity	Current	Dijkstra et al. [44] ^1^	Taylor et al. [17] ^2^
Sitting	89.2	76.8	85.2
Lying	83.1	64.6	98.0
Upright ^1^	93.1	80.2	56.1

^1^ Dijkstra et al. and Taylor et al. reported standing bouts. ^2^ Taylor et al. reported overall agreement, which was the total duration that the video observation and the accelerometer corresponded for each activity divided by the total duration each activity was observed on video.

## Data Availability

Data will be available upon reasonable request by contacting the principal investigator of the study, Professor Ngaire Kerse.

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
