# Peer review of "Validation of an Algorithm for Measurement of Sedentary Behaviour in Community-Dwelling Older Adults"

_sensors, 2023, doi:10.3390/s23104605_

Round 1

Reviewer 1 Report

Sensors (ISSN 1424-8220)

The following is an overview of the article Validation of an algorithm for measurement of sedentary behaviour in community-dwelling older adults (sensors-2363697). In this study, the author(s) proposes this paper investigated the ability of a semi-personalized algorithm to identify SB 357 and discriminate sitting, lying, and upright activities. The manuscript has contributions to the area of sedentary behavior; validation; older adults; wearable devices, digital health,

However, some points must be highlighted so that the author(s) can review and submit in another round of review: The following corrections are considered to be beneficial for the strengthening of the article.

1. The Conclusions should be reviewed again. The original aspect of the study and its difference from other studies should be clearly explained. (The conclusion should be explored better and it needs to contemplate the eventual restrictions of the developed technique to address future works in this area.)

2. The abstract must be make strong. Abstract should be reviewed again.

3. Some sentences have spelling errors. (Punctuation marks, spaces, etc.). Some places should be left space.

4. It has been a comprehensive study in the literature in recent years. If there are more current literature studies, these should be examined in detail and added to the literature section (Especially, sedentary behaviour; validation; older adults; wearable device, digital  health, activity recognition studies.). For example;

Wearable device use in older adults associated with physical activity guideline recommendations: Empirical research quantitative. Journal of Clinical Nursing.

A new approach for physical human activity recognition based on co-occurrence matrices. The Journal of Supercomputing, 78(1), 1048-1070.

Consumer wearable device-based measures of physical activity and energy expenditure in community-dwelling older adults with different levels of frailty: A STROBE compliant study. Medicine, 101(52), e31863.

A new approach for physical human activity recognition from sensor signals based on motif patterns and long-short term memory. Biomedical Signal Processing and Control, 78, 103963.

5. The authors should compare the results of their method with those of previous studies. As mentioned in the literature, there are several methods with very high accuracy, even better than the proposed method. Author(s) can do compare table (A new table can add about previous studies to result section.). This subject is very important.

6. The motivations of the proposed method are not clear. Which problem does the proposed method attempt to solve? Why the other existing diagnosis methods failed to solve it? What are the advantages of the proposed method compared to other methods? Those should be illustrated more clearly.

7. Carefully check all grammatical error. Still, the English language should be improved. I suggest asking for help from a native English

I think it is ACCEPTABLE after the MAJOR Revisions mentioned.

English language should be improved.

Reviewer 2 Report

I suggest to rewrite the abstract, it reports too much "numbers" instead to give a qualitative presentation of the results and the methods,

I would like a deeper discussion on the methods used to find the correct threshold  and how the authors avoid the introduction of biases. 

Reviewer 3 Report

Line 109: Sampling frequency of a video camera can be measured in FPS (Frames per seconds) instead Hz.

Line 110: AX6 and tablet were connected to a laptop placed on participant's home via Bluetooth? And the laptop is transmitting information further via the website? It's not clear what is the pipeline of the information transmission.

Table 2: can be written as a supplementary material(s). A flow diagram can show better the signal processing pipeline.

2.3.5 Data analysis: Algorithm flow is weakly described. How did you performed the training step? What ML or AI models have been used for the final algorithm?

Results: Since you test algorithm performance as correct time SB, you shall include and a chart/table results about the performance of different SB distinction. E.g. in 8/10 cases algorithm identified standing position.

Figure 2: size font can be increased. Hard to read in normal mode without zooming the content.

Conclusion: What would be the real application of the developed wearable device and algorithm? The purpose was only to train an algorithm to obtain a software solution or to obtain a hardware-software device?

Round 2

Reviewer 1 Report

Sensors (ISSN 1424-8220) | An Open Access Journal from MDPI

Dear Editor;

The author(s) made all the corrections mentioned (Sensors-2363697 - Validation of an algorithm for measurement of sedentary behaviour in community-dwelling older adults).

The length of the paper is enough in terms of a scientific paper. Considering studies conducted and results obtained, it is believed that the paper is eligible to be published in your journal after your approval.

I think it is ACCEPTABLE  in your journal after your approval as editor.